# MODEL-DIFF: A TOOL FOR COMPARATIVE STUDY OF LANGUAGE MODELS IN THE INPUT SPACE

## ABSTRACT

Comparing two (large) language models (LMs) side-by-side and pinpointing their prediction similarities and differences on the same set of inputs are crucial in many real-world scenarios, e.g., one can test if a licensed model was potentially plagiarized by another. Traditional analysis compares the LMs' outputs on some benchmark datasets, which only cover a limited number of inputs of designed perspectives for the intended applications. The benchmark datasets cannot prepare data to cover the test cases from *unforeseen* perspectives which can help us understand differences between models unbiasedly. In this paper, we propose a new model comparative analysis setting that considers a large input space where brute-force enumeration would be infeasible. The input space can be simply defined as all token sequences that a LM would produce low perplexity on — we follow this definition in the paper as it would produce the most human-readable inputs. We propose a novel framework Model-diff that uses text generation by sampling and deweights the histogram of sampling statistics to estimate prediction differences between two LMs in this input space efficiently and unbiasedly. Model-diff achieves this by drawing and counting the inputs at each prediction difference value in negative log-likelihood. Experiments reveal for the first time the quantitative prediction differences between LMs in a large input space, potentially facilitating the model analysis for applications such as model plagiarism.

## 1 INTRODUCTION

It is crucial in many real-world scenarios to compare two (large) language models (LMs) side-by-side and pinpoint their prediction differences. For example, the prediction differences may help identify which model agrees more with human annotations (Liu et al., 2020; Hendrycks & Gimpel, 2016; Hendrycks et al., 2019; Hsu et al., 2020; Lee et al., 2017; 2018; Liang et al., 2018; Mohseni et al., 2020; Ren et al., 2019; Szegedy et al., 2013; Rozsa et al., 2016; Miyato et al., 2018; Kurakin et al., 2016; Xie et al., 2019; Madry et al., 2017); it can also identify model plagiarism between a licensed open-sourced model and its small variate version whose weights are added small noise (PrimerYang). Traditional analysis compares the LMs' outputs on the same benchmark datasets which only cover a limited number of inputs from their designed perspectives. However, (large) LMs have recently been deployed online and widely accessible to the public. As users in principle can input any types of data that could potentially cause the models to behave unexpectedly, it could be beneficial if we can compare models by a large amount of data without favoring the designed perspectives. The challenge of using benchmark datasets in this case is twofold: (1) the tested perspectives are limited by the types of test sets, and (2) the variety of inputs from the same perspective is limited by the dataset size.

In this paper, we propose to compare models with a large number of data, or more generally, in a (discrete) input space that is finite but computationally impossible to enumerate all inputs. While the combination of all token sequences is a straightforward space, it is not ideal as most of them are sequences with random tokens, hence not beneficial for analysis. One reasonable input space is the collection of human-understandable inputs. It can be defined as all token sequences that an LM produces negative log-likelihood (NLL)[1] within a range of small values. Here we do not treat models as language generator but as language modeler, where a (good) language model is expected

---

[1]NLL is log-probability or log-perplexity which is the loss to train next token prediction for LMs.

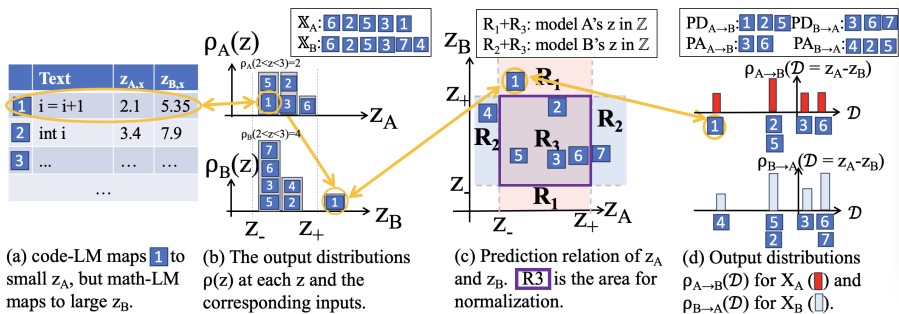

(a) code-LM maps 1 to small $z_A$, but math-LM maps to large $z_B$.

(b) The output distributions $\rho(z)$ at each z and the corresponding inputs.

(c) Prediction relation of $z_A$ and $z_B$. R3 is the area for normalization.

(d) Output distributions $\rho_{A \to B}(\mathcal{D})$ for $X_A$ ( ) and $\rho_{B \to A}(\mathcal{D})$ for $X_B$ ( ).

Figure 1: Overview of Model-diff with hypothetical models, code-LM (model $A$) and math-LM (model $B$). (a) code-LM assigns "i=i+1" (circled in orange ○) a small output value (z=NLL) but math-LM assigns a large NLL. (b) The set of inputs $\mathbb{X}_A$ ($\mathbb{X}_B$) that model $A$ ($B$) maps to a predefined output range $\mathbb{Z} = [z_-, z_+]$. The Output distributions $\rho_A(z)$ and $\rho_B(z)$ are the count of inputs at each $z$. (c) Using each model's prediction $z_{A,\mathbf{x}}$ and $z_{B,\mathbf{x}}$ as two coordinate axes indicates the relation of models' predictions of the same input (e.g. $z_{A,\mathbf{x}} < z_{B,\mathbf{x}}$ for "i=i+1"). The number of inputs in R3 is used to normalize statistics for sampling (Sec. 3.3). (d) Feeding the inputs in $\mathbb{X}_A$ to both models, compute each $\mathcal{D} = z_{A,\mathbf{x}} - z_{B,\mathbf{x}}$, and count the number of inputs to get $\rho_{A \to B}(\mathcal{D})$ (red histogram). Repeat for model $B$ to get $\rho_{B \to A}(\mathcal{D})$ (light blue histogram). The "i=i+1" is mapped to a very negative $\mathcal{D}$ value.

to output low NLL (low perplexity) to inputs of text sequences that humans see as reasonable and fluent, and high NLL (high perplexity) to inputs that are nonsensical or unreasonable. We focus on this input space because the number of human-understandable sequences is large enough to cover the unforeseen perspectives and data to analyze the models.

To tackle this new evaluation of a large input space, we propose a sampling-based comparative analysis framework, Model-diff, that can efficiently estimate the prediction difference for the *types* and *count* of the inputs at each level of the prediction difference value. Consider two hypothetical models that optimize the NLL loss. They are fine-tuned to domain tasks as code-LM (model $M_A$) and math-LM (model $M_B$). In Fig 1(a), the types of inputs that code-LM predicts with a low NLL could be the variable assignment code such as "i=i+1", whereas math-LM thinks this text is a wrong math equation and thus assigns a high NLL (Fig 1(b)). If two models predict similar NLLs for each input in an input space, the count of prediction difference (e.g. $\mathcal{D} = \text{NLL}_A - \text{NLL}_B$) concentrates around 0, indicating the models are similar. If not, we can quantify the difference by the number of inputs at each $\mathcal{D}$. Because the input space is not enumerable, we sample the input space to assess the prediction differences between the two models by scrutinizing the inputs and counting the number of inputs given different output values (e.g., there are 10 math equations when $\mathcal{D} = -5$ and 23 code inputs when output is $\mathcal{D} = 10$ etc), similar to Fig 1(d).

Model-diff samples the models whose predictions are within a range of low NLL. The outputs of sampling, including the sampled inputs and the count histogram that is processed to get the *output distribution*, can be used to compare the types and counts of the inputs. Output distribution (Liu et al., 2023b) is a distribution of the count of the inputs given each output value, the exact quantity for count comparison. Comparing the summations of two models over this quantity for differentvalues can help compare the total *count* of inputs in an input space even though the input space is usually too large to be enumerated. Scrutinizing the sampled inputs for different outputs can help understand the *types* of the inputs. Model-diff leverages them to quantify the agreed/disagreed predictions between the two models. Moreover, to ensure a fair comparison, each model will propose its own input space containing inputs of low NLLs and compare predictions of the other model. Model-diff involves a novel normalization strategy to normalize the two output distributions sampled from the input spaces proposed by each model.

**Contributions**

- We propose a new comparative analysis setting between two models by examining their prediction differences on the full input space, in contrary to leveraging crafted datasets of testing foreseen perspectives.

- To address the infeasible compute time of enumeration, we propose Model-diff. It can help understand the *type(s)* and the (relative) *count* of the agreed/disagreed predictions between two models in the meaningful input spaces.
- We confirm the correctness of Model-diff through a Toy example. Further experiments show Model-diff can find prediction differences for GPT2 with various sequence lengths and Llama. Moreover, application to model plagiarism with Model-diff discovers distinctive patterns for a model whose weights are added noise. This could be a useful signal for further confirmation of plagiarism.

## 2 THE MODEL-DIFF FRAMEWORK

Model-diff leverages the output distribution at each prediction difference $\mathcal{D}$ and the corresponding inputs mapped to $\mathcal{D}$ to analyze the types and count of the inputs at each $\mathcal{D}$ value. We first introduce the concept of output distribution and then how we use it for prediction difference analysis in an ideal case when enumeration is available. Next section we will discuss how to derive the quantities needed for this analysis when enumeration is replaced by sampling.

**Background: Output distribution.** Given the entire discrete input space $\Omega = \{0, ..., M\}^N$ and a training set $\Omega_T \subseteq \Omega$, a model $f(\mathbf{x})$ learns to map inputs $\mathbf{x} \in \Omega_T$ to output $z \in \mathbb{R}$. As the current language models (LMs) are trained to predict the next token, we choose the loss function, negative log-likelihood (NLL), as the output. Later we also define output distribution for the parameter of prediction difference $\mathcal{D}$. Each input $\mathbf{x}$ is a sequence of $N$ tokens. $M + 1$ is the vocabulary size. Each of the $N$ tokens takes one of the $M + 1$ words. The **output distribution** in an input space $\Omega^*$ is the distribution of the count for each $z$. $\Omega^*$ can be $\Omega$ or some other space $\Omega_M$ specified by a generative model M. As every input in $\Omega^*$ holds equal importance for analysis, the inputs within $\Omega^*$ should follow the principle of equal *a priori* probabilities – each input within $\Omega^*$ follows a uniform distribution. Mathematically, the output distribution $\rho(z)$ is defined as:

$$\rho(z) = \sum_{\mathbf{x} \in \Omega^*} \delta(z - f(\mathbf{x})),$$

where $\delta(\cdot)$ is 1 if the input $z - f(\mathbf{x}) = 0$, or $\delta(\cdot)$ is 0 otherwise. In practice, a histogram is used to collect the statistics (the y-axis is the count and the x-axis is the output values $z$). The sampled inputs with similar output values in a small range $[z - \Delta z, z + \Delta z]$ are called **representative inputs** at $z$ and are mapped to the same $z$ bin. $\Delta z$ is a small positive constant.

### 2.1 MODEL-DIFF FRAMEWORK

**Introductory example and goal.** Assume we have infinite computing power to enumerate the inputs in a large input space to get the ground truth statistics. Because the inputs with very low NLL are repetitive sequences that are not understandable by humans (Holtzman et al., 2019) whereas inputs with (slightly) higher NLL are human understandable, we avoid the input space that favors the inputs with very low NLL or contains the inputs of with a specific NLL. Instead, we flexibly define a range of low (NLL) output values $\mathbb{Z} = [z_-, z_+]$ and treat the inputs whose output values in $\mathbb{Z}$ as equally important[2]. Thus, our comparative analysis is w.r.t. the chosen $\mathbb{Z}$. In Fig 1 (b), there is an input space $\Omega$ with a large number of inputs where model $M_A$ maps 5 inputs to outputs within $\mathbb{Z}$ and Model $M_B$ maps 6 inputs within $\mathbb{Z}$. These 5 (and 6) inputs mapped by $M_A$ ($M_B$) form a set called $\mathbb{X}_A$ ($\mathbb{X}_B$). Some of the input(s) from $\mathbb{X}_A$ may be predicted with higher (or lower) $z$ by $M_B$ than $M_A$ predicts, such as the circled input ($\bigcirc$). Model-diff's comparative analysis aims to find the types and counts of these inputs that are predicted with different output values by the two models.

**Comparative analysis with $\rho_{A \to B}(\mathcal{D})$.** Define $A \to B$ as the representative inputs $\mathbb{X}_A$ from model $M_A$ are evaluated by model $M_B$, and $B \to A$ is vice versa. It is more convenient to consider the prediction relation in Fig. 1(c) in terms of $\mathcal{D}$. In Fig. 1(d), Model-diff uses the output distributions $\rho_{A \to B}(\mathcal{D})$ and $\rho_{B \to A}(\mathcal{D})$ for comparative analysis, where $z_{A,\mathbf{x}} = M_A(\mathbf{x})$, $z_{B,\mathbf{x}} = M_B(\mathbf{x})$, and $\mathcal{D}$ is the *predictive difference* $\mathcal{D} = d(z_{A,\mathbf{x}}, z_{B,\mathbf{x}})$ for the same input $\mathbf{x}$. $d(\cdot)$ is a measurement of difference. $\rho_{A \to B}(\mathcal{D})$ is a distribution of the total count for (all) inputs corresponding to meaningful

---

[2]Considering the inputs whose outputs within $\mathbb{Z}$ is helpful to analyze more human-understandable inputs instead of focusing on the inputs with (very) low NLLs, but our analysis works in other input space.

output values $\mathbb{Z}$ for model $M_A$ at each value $\mathcal{D}$. Intuitively, the larger $|\mathcal{D}|$ means the two models' predictions are more different for input $\mathbf{x}$ and the larger $\rho_{A \to B}(\mathcal{D})$ means a larger number of inputs whose output differences are by $\mathcal{D}$. $\rho_{B \to A}(\mathcal{D})$ works similarly. Our setting and experiments focus on LMs and thus we use the difference in NLL between two models as $\mathcal{D}$: $\mathcal{D} = \text{NLL}_{A,\mathbf{x}} - \text{NLL}_{B,\mathbf{x}}$. Other output and $d(\cdot)$ can be used for different applications. $\mathcal{D}$'s output distributions for $\mathbb{X}_A$ and $\mathbb{X}_B$ are:

$$\rho_{A \to B}(\mathcal{D}) = \sum_{\mathbf{x} \in \mathbb{X}_A} \delta(\mathcal{D} - (z_{A,\mathbf{x}} - z_{B,\mathbf{x}})), \tag{1}$$

$$\rho_{B \to A}(\mathcal{D}) = \sum_{\mathbf{x} \in \mathbb{X}_B} \delta(\mathcal{D} - (z_{A,\mathbf{x}} - z_{B,\mathbf{x}})), \tag{2}$$

where $\delta(\cdot)$ is 1 if the input $\mathcal{D} - (z_{A,\mathbf{x}} - z_{B,\mathbf{x}}) = 0$, or $\delta(\cdot)$ is 0 otherwise.

Define a varying threshold $\lambda$ for $\mathcal{D}$ values. We can get the following comparative analysis as illustrated in Fig. 1(d):

- Prediction disagreement (**PD**) $\text{PD}_{A \to B} = \sum_{\mathcal{D} < \lambda \leq 0} \rho_{A \to B}(\mathcal{D})$ is the amount of model $A$'s representative inputs $\mathbb{X}_A$ that model $B$ assigns with **higher** NLL: $z_{B,\mathbf{x}} > z_{A,\mathbf{x}}$ (e.g. the ■ with labels "1" (circled in ○), "2" and "5").
- Prediction agreement (**PA**) $\text{PA}_{A \to B} = \sum_{\mathcal{D} > \lambda \geq 0} \rho_{A \to B}(\mathcal{D})$ is the amount of model $A$'s representative inputs $\mathbb{X}_A$ that model $B$ assigns with **lower** NLL: $z_{B,\mathbf{x}} < z_{A,\mathbf{x}}$ (e.g. the ■ with labels "3" and "6").
- $\text{PA}_{B \to A} = \sum_{\mathcal{D} < \lambda \leq 0} \rho_{B \to A}(\mathcal{D})$ is the amount of model $B$'s representative inputs $\mathbb{X}_B$ that model $A$ assigns with **lower** NLL: $z_{A,\mathbf{x}} < z_{B,\mathbf{x}}$ (e.g. the ■ with labels "4", "2", and "5").
- $\text{PD}_{B \to A} = \sum_{\mathcal{D} > \lambda \geq 0} \rho_{B \to A}(\mathcal{D})$ is the amount of model $B$'s representative inputs $\mathbb{X}_B$ that model $A$ assigns with **higher** NLL: $z_{A,\mathbf{x}} > z_{B,\mathbf{x}}$ (e.g. the ■ with labels "3", "6", and "7").

Therefore, the ratio of their count is:

$$\text{PD}_{A \to B} : \text{PA}_{A \to B} : \text{PA}_{B \to A} : \text{PD}_{B \to A}, \tag{3}$$

which is important in understanding the count of agreed/disagreed predictions. For example, Fig 1(d) shows the ratio is 3:2:3:3 when $\lambda = 0$. Moreover, by examining the representative inputs at each $\mathcal{D}$ value, we can gain insights into the types of inputs that the two models predict differently by $\mathcal{D}$.

## 2.2 Analysis with input annotations

**Agreement between model prediction difference and human annotations.** To understand which model agrees more with humans' annotations (Liu et al., 2023a), humans can annotate the representative input at each prediction difference $\mathcal{D}$. Humans annotate with score from 1 when a representative input agrees with the training objective ("perfectly good") to 0 otherwise ("completely bad"). The annotation score $r_A(\mathcal{D})$ is the average of all the annotated representative inputs for model $A$ at $\mathcal{D}$'s nearby values ($\mathcal{D} - \Delta\mathcal{D}, \mathcal{D} + \Delta\mathcal{D}$), where $\Delta\mathcal{D}$ is a small positive constant. Using $\text{PD}_{A \to B}(\mathcal{D})$ as an example (it could be one of the four terms in Equ. 9), the true positive at $\mathcal{D}$ is the proportion of "good" inputs times the count:$r_A(\mathcal{D})\rho_{A \to B}(\mathcal{D})$. Summing over $\mathcal{D} < \lambda \leq 0$, we get precision:

$$\text{precision} = \frac{\sum r_A(\mathcal{D})\rho_{A \to B}(\mathcal{D})}{\text{PD}_{A \to B}(\mathcal{D})}. \tag{4}$$

The recall is:

$$\text{recall} = \frac{\sum r_A(\mathcal{D})\rho_{A \to B}(\mathcal{D})}{\text{number of positive inputs}}$$
$$\propto \sum r_A(\mathcal{D})\rho_{A \to B}(\mathcal{D}), \tag{5}$$

because the number of positive inputs is a constant in $\mathbb{X}_A$. We can then use these two quantities to measure whether humans believe the prediction of higher NLL is reasonable. In the above example of prediction disagreement $\text{PD}_{A \to B}(\mathcal{D})$ on $A$'s representative inputs by $B$, if both precision and recall are low, then model $A$ maps a lot of "bad" inputs to low NLL and thus model $B$'s disagreement is reasonable.

## 3 EFFICIENT AND UNBIASED SAMPLING IN MODEL-DIFF

In reality, enumeration is impossible because of computation inefficiency. We need to estimate the key distributions $\rho_{A \to B}(\mathcal{D})$ and $\rho_{B \to A}(\mathcal{D})$ by sampling. We first introduce the background of text generation by sampling, and the method(s) to sample the output distribution. We then discuss how to acquire comparable $\rho_{A \to B}(\mathcal{D})$ and $\rho_{B \to A}(\mathcal{D})$ through output distribution and normalization.

### 3.1 BACKGROUND AND TERMINOLOGY

**Text Generation by Sampling.** Besides generating the next token in an autoregressive manner, sampling methods are common in text generation in language models (Kumar et al., 2022; Qin et al., 2022), by Markov Chain Monte-Carlo (MCMC). It starts with a sequence of random tokens and by tweaking the tokens randomly to lower the NLLs, a sequence of understandable text is generated. MCMC sampling is employed because enumeration of the input space in general is not possible. As pointed out (Du et al., 2023), text generation by sampling in principle should employ samplers of discrete input space (Goshvadi et al., 2024; Grathwohl et al., 2021; Zhang et al., 2022). These samplers sample the target distribution

$$p(\mathbf{x}) \propto \exp(g(\mathbf{x})/T), \tag{6}$$

where $g(\cdot)$ is called (negative) "energy" and $T$ is a predefined parameter (temperature). When $T$ is 1, $g(\cdot)$ is the log-probability which is popular in many machine learning problems when they learn to model log-probability (LeCun et al., 2006). $p(\mathbf{x})$ in Equ. 6 is a common target distribution for sampling in machine learning. Importantly, it biases the inputs with high $g(\mathbf{x})$.

Model-diff adopts the exact same sampling setting of discrete inputs and this is the major bottleneck of Model-diff. The time complexity of Model-diff is therefore similar to text generation by sampling. Post-processing of Model-diff after text generation by sampling only takes a few hours.

**Sampling the output distribution.** Parallel Tempering and Histogram Reweighting (PTHR) (Hukushima & Nemoto, 1996; Swendsen & Wang, 1986) is commonly used to sample output distribution. It starts with the results of text generation by sampling for target distribution of Equ. 6. Because the MCMC samplers sample $\mathbf{x}$ more often whose output $g(\mathbf{x})$ is larger, it needs reweights the sampled distributions by $\exp(\cdot)$ to acquire the output distribution. Therefore, sampling output distribution can generate the same statistics as if we were sampling uniformly the input space without biasing the inputs with large $g(\mathbf{x})$. Moreover, PTHR is a downstream task of text generation by sampling and it is compatible with MCMC samplers. Therefore it can take advantage of the development of MCMC samplers that follow the same target distribution of Equ. 6.

### 3.2 SAMPLING WITH PROBABILITY WEIGHTS OF $\tilde{\rho}_A(z)$ OR $\tilde{\rho}_B(z)$

For very large input space, exact values of $\rho_{A \to B}(\mathcal{D})$ and $\rho_{B \to A}(\mathcal{D})$ cannot be estimated because $\mathbb{X}_A$ and $\mathbb{X}_B$ (Sec. 2.1 Introductory example and goal) are not available as enumeration of the input space is infeasible. Thus, we need to estimate them by sampling. We denote the sampled quantity used for practical analysis with "Tilde" (e.g. $\tilde{\rho}$) in contrast to the quantity from ground truth enumeration without "Tilde" (e.g. $\rho$) for conceptual discussion purposes.

As mentioned in Sec. 2.1, we focus on the case where every input whose outputs within $\mathbb{Z}$ as equally important; therefore the outputs that contain more inputs should be sampled more often. Text generation by sampling is not directly applicable because it favors low NLL. We instead leverage the output distribution $\rho_A(z)$ (or $\rho_B(z)$) that describes the various numbers of inputs mapped to each output value by a model. For example, as shown in Fig. 1 (b), one output value of model $B$ has 4 inputs, and should be selected 2 times more frequently than the other output value with only 2 inputs. Output distribution $\rho_A(z)$ (or $\rho_B(z)$) ensures the sampled representative inputs follow the frequency of appearance for the different output values in $\mathbb{Z}$ for model $A$ (or $B$) result in a sampling process as if we were uniformly extracting inputs from $\mathbb{X}_A$ and $\mathbb{X}_B$.

In practice, we apply the well-established algorithms of text generation by sampling and PTHR. After text generation by sampling, we compute $\tilde{\rho}_A(z)$ and $\tilde{\rho}_B(z)$ that approximate $\rho_A(z)$ and $\rho_B(z)$ through PTHR. We then sample an output value $z$ with probability weights $\tilde{\rho}_A(z)$ (or $\tilde{\rho}_B(z)$) so that the output $z$ with more inputs will be sampled more often. Afterward, we uniformly choose an

input $\mathbf{x}$ whose output $M_A(\mathbf{x}) = z$ (or $M_B(\mathbf{x}) = z$). More math details about this sampling are in Appendix D.1. Many of these sampled $\mathbf{x}$ are fed to both models, compute their $\mathcal{D}$, and record the count in a histogram of output distributions which are the output of this process – un-normalized $\tilde{\rho}_{A \to B}(\mathcal{D})$ and $\tilde{\rho}_{B \to A}(\mathcal{D})$.

As an alternative, we can directly sample $\tilde{\rho}_{A \to B}(\mathcal{D})$ and $\tilde{\rho}_{B \to A}(\mathcal{D})$, but we find the two-stage sampling is more flexible – first sampling $\tilde{\rho}(z)$ for individual model and then $\tilde{\rho}(\mathcal{D})$ when we need to compare them – because $\tilde{\rho}(z)$ can be reused. This two stage formulation also leads to the correct results (Sec. 4.2). Moreover, if other input spaces are used, we can obtain un-normalized $\tilde{\rho}_{A \to B}(\mathcal{D})$ and $\tilde{\rho}_{B \to A}(\mathcal{D})$ easily, such as using the sampled inputs without $\tilde{\rho}_A(z)$ or $\tilde{\rho}_B(z)$.

### 3.3 NORMALIZATION

The sampled $\tilde{\rho}_{A \to B}(\mathcal{D})$ and $\tilde{\rho}_{B \to A}(\mathcal{D})$ are not comparable yet, because sampling needs normalization. Traditionally, we can normalize through the area under curve of the sampled histogram so the distribution is normalized to $1.0$. However, we are only interested in comparing the inputs whose NLLs are low and do not need to sample the inputs to cover all the output values. Thus, we develop a normalization method by using the area where both models predict within $\mathbb{Z}$ (R3 in Fig 1(c)).

To see how this works, we had the ground truth result by enumeration is 3:2:3:3 (Equ. 3) for Fig. 1(d), when $\lambda = 0$. On the other hand, we suppose enumeration is impossible. If we sample 100 inputs from $M_A$, around 80 of which are expected to be predicted within $\mathbb{Z}$ by both models (■ with "2","5","3","6" are in $\mathbb{Z}$, but "1" is not.). Among these 100 inputs, 60 of them are $\text{PD}_{A \to B}$ and 40 of them are $\text{PA}_{A \to B}$. We can repeat this process when we sample 300 inputs by model $M_B$ and 200 of them are from $\mathbb{Z}$ by $M_A$. Among these 300 inputs, 150 of them are $\text{PD}_{B \to A}$ and 150 of them are $\text{PA}_{B \to A}$. We can use the two sets of the sampled inputs that are commonly predicted by the two models within $\mathbb{Z}$ as the denominators (80 for $M_A$ and 200 for $M_B$) to fix Equ. 3. This allows us to compare the sum of the following output distributions after being divided by denominators:

$$\sum \rho_{A \to B}(\mathcal{D}) \propto \frac{\sum \tilde{\rho}_{A \to B}(\mathcal{D})}{|\tilde{\mathbb{X}}_{A \to B}|} \ , \tag{7}$$

$$\sum \rho_{B \to A}(\mathcal{D}) \propto \frac{\sum \tilde{\rho}_{B \to A}(\mathcal{D})}{|\tilde{\mathbb{X}}_{B \to A}|} \ , \tag{8}$$

where the proportions have the same weight (validation of this normalization is in Appendix D.2). $\tilde{\mathbb{X}}_{A \to B}$ ($|\tilde{\mathbb{X}}_{A \to B}|$=80 in the above example) is the set of the inputs sampled by model $M_A$ within $\mathbb{Z}$ and model $M_B$ also predicts within $\mathbb{Z}$, and $\tilde{\mathbb{X}}_{B \to A}$ ($|\tilde{\mathbb{X}}_{B \to A}| = 200$) is vice versa. Therefore, by considering Equ. 7 and Equ. 8, Equ. 3 becomes the normalized ratio:

$$\frac{\sum\limits_{\mathcal{D} < \lambda \le 0} \tilde{\rho}_{A \to B}(\mathcal{D})}{|\tilde{\mathbb{X}}_{A \to B}|} : \frac{\sum\limits_{\mathcal{D} > \lambda \ge 0} \tilde{\rho}_{A \to B}(\mathcal{D})}{|\tilde{\mathbb{X}}_{A \to B}|} : \frac{\sum\limits_{\mathcal{D} < \lambda \le 0} \tilde{\rho}_{B \to A}(\mathcal{D})}{|\tilde{\mathbb{X}}_{B \to A}|} : \frac{\sum\limits_{\mathcal{D} > \lambda \ge 0} \tilde{\rho}_{B \to A}(\mathcal{D})}{|\tilde{\mathbb{X}}_{B \to A}|} \tag{9}$$

The ratio Equ. 9 of the example is $\frac{60}{80} : \frac{40}{80} : \frac{150}{200} : \frac{150}{200}$, the same as the ground truth ratio. In summary, the sampled statistics with this normalization (Equ. 7 and 8) reflect the ground truth and they are comparable. Finally, a new model $C$ with the same training target may need to be compared with $A$ and $B$. We derive the relation between models $B$ and $C$ when they are compared with model $A$. The details are in Appendix E.

**Model-diff pipeline.** In practice, Model-diff consists of four steps:

(a) Use text generation by sampling (Sec. 3.1) to generate inputs, collect the statistics (frequency histogram), and use PTHR to compute output distribution $\tilde{\rho}_A(z)$ for model $A$'s meaningful output values $\mathbb{Z}$.
(b) Sample the collected representative inputs from (a) with weights $\tilde{\rho}_A(z)$ (Sec. 3.2). Feed each sampled input from $A$ to model $B$ to compute prediction (output) difference $\mathcal{D}$. The $\mathcal{D}$ of the sampled inputs from $A$ forms a distribution $\tilde{\rho}_{A \to B}(\mathcal{D})$ of prediction difference. It is normalized (Sec. 3.3) to get a comparable $\rho_{A \to B}(\mathcal{D})$.
(c) We repeat the same process to get $\tilde{\rho}_{B \to A}(\mathcal{D})$ for model $M_B$.
(d) $\rho_{A \to B}(\mathcal{D})$, $\rho_{B \to A}(\mathcal{D})$, and the correspondent sampled inputs are compared and analyzed to quantify prediction difference (Sec. 2.1 Analysis with output distribution), sometimes with input annotations (Sec. 2.2).

## 4 EXPERIMENTS

We first apply Model-diff to a Toy example where the enumeration of all inputs is affordable to confirm Model-diff's correctness (Sec. 4.2). We then apply it to two pretrained GPT2 models (Radford et al., 2019) with various sequence lengths (25 and 100) and Llama models (Touvron et al., 2023a;b) with sequence length 25 (Sec. 4.3). This shows Model-diff is applicable to real-world models. As we have confirmed the applicability of Model-diff, we will show the types and count Model-diff focuses on can be useful in real-world applications (Sec. 4.4).

### 4.1 EXPERIMENTAL SETTINGS

Our sampling target is the negative-log-likelihood (NLL), the training loss used for next-token predictions. Though a low NLL generally indicates the model (strongly) believes the input is close to the training distribution, the inputs with very low NLLs are repeating words that are incomprehensible by humans (Appendix G and Holtzman et al. (2019)). Therefore, we generally set a reasonable range of $\mathbb{Z}$ and only consider inputs whose NLL $\in \mathbb{Z}$. We choose the range $\mathbb{Z}$ by ensuring the bins have human-understandable inputs, but our method works in any range (input space) selected for the specific task. Fig. 2 shows the sampling results of $\mathcal{D} = \text{NLL}_A - \text{NLL}_B$ with one unit of standard deviation for three runs (two for Toy) after we have the PTHR results. Tab. 1 shows the detailed statistics about Fig. 2 for Model-diff analysis. More details of experimental settings are in Appendix F.

### 4.2 TOY EXAMPLE

**Toy** is a simple experiment with dataset of sequences $\{\mathbf{x}^{(i)}\}$ with length 8. Each token $x_j$ for an input $\mathbf{x}^{(i)}$ is an integer from 0 to 9 inclusive; vocabulary size is 10. The entire input space is $10^8$ which is enumerable. The training objective is $(\sum x_j) \bmod 30 = 0$. The **GPT2-small-Toy** has 4 heads and 6 layers. The **GPT2-large-Toy** has 8 heads and 8 layers. Both models can generate sequences that satisfy the objective with $100.0\%$ after training.

**Analysis.** Fig. 2(a) shows the output distribution $\tilde{\rho}(\mathcal{D})$, where we set $\mathcal{D} = \text{NLL}_{\text{small}} - \text{NLL}_{\text{large}}$. Exp.1 in Tab. 1 shows the statistics of Fig. 2(a). $\tilde{\rho}(\mathcal{D})$ on GPT2-small-Toy's representative inputs ranges from $-0.9$ to $0.25$, indicating that GPT2-large-Toy's predicted NLL on some GPT2-small-Toy's representative inputs can be up to 0.9 larger and 0.25 smaller on some other inputs than GPT2-small-Toy predicts. On the other hand, $\tilde{\rho}(\mathcal{D})$ on GPT2-large-Toy ranges from $-0.35$ to $0.55$, indicating GPT2-small-Toy's predicted NLL on some GPT2-large-Toy's representative inputs can be up to 0.55 larger on some inputs but 0.35 smaller on some other inputs than GPT2-large-Toy predicts. Comparison of prediction disagreements between GPT2-small-Toy's $\min \mathcal{D}$ ($-0.9$) and GPT2-large-Toy's $\max \mathcal{D}$ (0.55) shows GPT2-large-Toy disagrees more strongly on GPT2-small-Toy's representative inputs than GPT2-small-Toy disagrees on GPT2-large-Toy's representative inputs.

In terms of the number of prediction disagreement/agreement, the normalized ratio of count is $1.0 : 0.75 : 0.55 : 0.75$. Compared to $\text{PD}_{\text{s(mall)} \to \text{l(arge)}}$ (1.0), $\text{PA}_{\text{s} \to \text{l}}$ means 0.75 amount of GPT2-small-Toy's representative inputs would be assigned with lower NLL by GPT2-large-Toy, and $\text{PA}_{\text{l} \to \text{s}}$ means 0.55 amount of GPT2-large-Toy's representative inputs would be predicted with lower NLL by GPT2-small-Toy. Lastly, compared to $\text{PD}_{\text{s} \to \text{l}}$, $\text{PD}_{\text{l} \to \text{s}}$ means 0.75 amount of GPT2-large-Toy's representative inputs would be predicted with higher NLL by GPT2-small-Toy. Model-diff shows the two models have a high overlap of predictions as the $\mathcal{D}$ concentrates around 0.

**Correctness of Model-diff.** We enumerate all the sequences as the ground truth in Fig 2(a). The ground-truth plot is closely aligned with our sampled plot. Lastly, our sampled ratio is very close to the ground truth enumeration ratio $1.0 : 0.72 : 0.56 : 0.73$. The toy example confirms the correctness of Model-diff where the sampling results can properly represent the enumeration; we can apply it to more complicated applications with confidence. Moreover, we use MCMC with a temperature equals to 1.0 to sample the GPT2-large-Toy. Fig. 6 shows that simple text generation by MCMC sampling does not lead to the same ground truth distribution for a range of output values (see Sec. 3.2), because it biases the inputs with low NLLs.

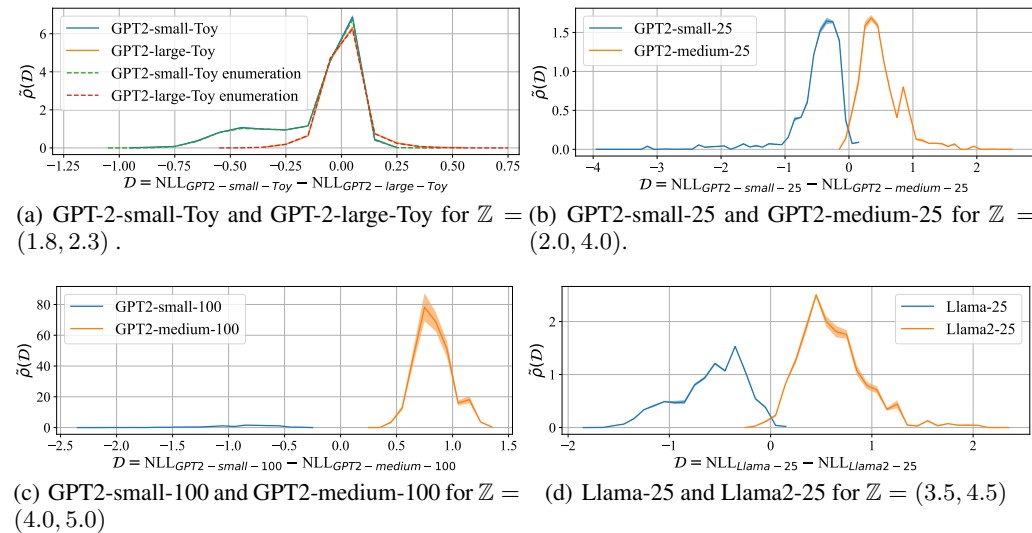

(a) GPT-2-small-Toy and GPT-2-large-Toy for $\mathbb{Z} = (1.8, 2.3)$ .

(b) GPT2-small-25 and GPT2-medium-25 for $\mathbb{Z} = (2.0, 4.0)$.

(c) GPT2-small-100 and GPT2-medium-100 for $\mathbb{Z} = (4.0, 5.0)$

(d) Llama-25 and Llama2-25 for $\mathbb{Z} = (3.5, 4.5)$

Figure 2: Comparing different language models using Model-diff on different input spaces. Except for (a), all the comparisons are done in the input space that a model believes to be reasonable human inputs by $\mathbb{Z}$.

| Exp | Repre Inputs from model A or B | $\mathcal{D}$ min | $\mathcal{D}$ max | $\text{PD}_{A \to B} : \text{PA}_{A \to B} : \text{PA}_{B \to A} : \text{PD}_{B \to A}$ (Equ. 9 with $\lambda = 0$) |
|---|---|---|---|---|
| 1 | A: GPT2-small-Toy | -0.9 | 0.25 | $1.00 : 0.75(\pm 0.00) : 0.55(\pm 0.01) : 0.75(\pm 0.00)$ |
|   | B: GPT2-large-Toy | -0.35 | 0.55 | |
| 2 | A: GPT2-small-25 | -3.95 | 0.15 | $1.00 : 0.02(\pm 0.00) : 0.01(\pm 0.00) : 1.03(\pm 0.02)$ |
|   | B: GPT2-medium-25 | -0.15 | 2.55 | |
| 3 | A:GPT2-small-100 | -2.25 | -0.25 | $1.00 : 0.00(\pm 0.00) : 0.00(\pm 0.00) : 29.84(\pm 3.14)$ |
|   | B: GPT2-medium-100 | 0.25 | 1.25 | |
| 4 | A: Llama-25 | -1.85 | 0.15 | $1.00 : 0.01(\pm 0.00) : 0.01(\pm 0.00) : 1.61(\pm 0.07)$ |
|   | B: Llama2-25 | -0.15 | 2.35 | |

Table 1: $\mathcal{D} = \text{NLL}_A - \text{NLL}_B$. Equ. 3 and 9 are normalized by the first term $\text{PD}_{A \to B}$; thus, the first term is 1.0. Our experiments will also set $\lambda = 0$. Besides $\lambda = 0$, other $\lambda$ values could be computed and analyzed similarly.

### 4.3 REAL-WORLD LANGUAGE MODELS

We apply Model-diff to two pretrained GPT2 models, GPT2-small and GPT2-medium with $\mathcal{D} = \text{NLL}_{\text{small}} - \text{NLL}_{\text{medium}}$. **GPT2-small-25** and **GPT2-medium-25** sample 25 tokens with GPT2-medium and with GPT2-small respectively. For longer sequence length, **GPT2-small-100** and **GPT2-medium-100** sample 100 tokens with GPT2-small and with GPT2-medium respectively.

Fig. 2(b) shows the $\tilde{\rho}(\mathcal{D})$ for both models with sequence length 25. In Exp.2 of Tab. 1, comparison between GPT2-small-25's $\min \mathcal{D}$ ($-3.95$) and GPT2-medium-25's $\max \mathcal{D}$ (2.55) shows GPT2-medium-25 disagrees more strongly on some GPT2-small-25's representative inputs than GPT2-small-25 disagrees on some GPT2-medium-25's representative inputs. However, the count ratio (Equ. 9) on Tab. 1 (Exp 2) shows the number of inputs for prediction agreements (0.02 vs 0.01) and prediction disagreements (1.0 vs 1.03) are almost the same for both models.

Moreover, the experiment on 100 sequence length in Fig. 2(c) and its statistics (Table. 1 Exp 3) show GPT2-small-100 and GPT2-medium-100 have distinctive characteristics. GPT2-small-100's $\min \mathcal{D}$ ($-2.25$) is almost two times larger than GPT2-medium-100's $\max \mathcal{D}$ (1.25) in absolute value, indicating the GPT2-medium-100 disagrees more strongly on some GPT2-small-100's representative inputs than vice versa. In terms count, $\text{PD}_{\text{m(edium)} \to \text{s(mall)}}$ is 29.84 times larger than $\text{PD}_{\text{s} \to \text{m}}$ (1.0). Lastly, the prediction agreement on each other's representative inputs is (extremely) low compared to prediction agreement.

**Model-diff on large language models.** We apply Model-diff to pre-trained Llama-7B[3] and Llama2-7B for sequence length 25 as **Llama-25** and **Llama2-25**. We use $\mathcal{D} = \text{NLL}_{\text{Llama}} - \text{NLL}_{\text{Llama2}}$.

Comparison between $\mathcal{D}$'s maximum for Llama2-25 (2.35) and minimum for Llama-25 ($-1.85$) in Fig. 2(d) and its statistics (Table 1 Exp 4) shows that Llama-25 disagrees more strongly on Llama2-25's representative inputs than vice versa. Moreover, Table 1 Exp 4 shows the count ratio of PA and PD. Compared to $\text{PD}_{\text{L(lama)}\rightarrow\text{(Llama)2}}$, the very low $\text{PA}_{\text{L}\rightarrow 2}$ and $\text{PA}_{2\rightarrow\text{L}}$ (both 0.01) show prediction agreement between the two models is low compared to $\text{PD}_{\text{L}\rightarrow 2}$ (1.0). But $\text{PD}_{2\rightarrow\text{L}}$ is around 1.6 times larger than the $\text{PD}_{\text{L}\rightarrow 2}$.

**Discussion.** We can further analyze the representative inputs corresponding to different $\mathcal{D}$. For example, in the experiment for GPT2-small-25 and GPT2-medium-25, we choose to inspect the inputs corresponding to large $|\mathcal{D}|$. Interestingly, we find that GPT2-medium-25 disagrees with GPT2-small-25 on the database inputs, whereas GPT2-small-25 disagrees with GPT2-medium-25 on inputs about computer media decoder and PCIe (see Appendix H).

Our results show Model-diff can quantitatively compare two models' low NLL input spaces in terms of count and types of the inputs. Moreover, the models with high capacity (more weights and/or more complex architectures) generally have a larger amount of representative inputs mapped to low NLL values. Notably, this does not mean they are more tolerant of the representative inputs of other models with lower capacity. They generally disagree more on the representative inputs from another model with lower capacity and the disagreement can be quantified by Model-diff.

### 4.4 APPLICATIONS

We demonstrate a few real-world examples of applications using the types and counts Model-diff focuses, as we have confirmed the applicability of Model-diff.

**Deciding which model is better.** We define our task of which model is better in terms of *which model's prediction agrees more with human annotation*. We achieve this by annotating the inputs. We choose to annotate the inputs from -1 to -0.6 and from 1 to 0.6 where the dominant number of inputs concentrates and $|\mathcal{D}|$ is not too small when the two models do not show significant prediction differences. We sum from the -1 to -0.6 for $\text{PD}_{small\rightarrow medium}$ and from 1 to 0.6 for $\text{PD}_{medium\rightarrow small}$. Using Equ. 4 and Equ. 5, we compute $0.56$ precision and recall is $0.32$ for GPT2-small-25. For GPT2-medium-25, we compute $0.58$ precision and recall is $0.57$. This shows while the two models disagree with the prediction of the other model's representative inputs, GPT2-medium-25's disagreement aligns more closely to human annotation. Therefore, GPT2-medium is a better model as its prediction agrees more with human annotation. Without introducing extra biases from datasets, we can use Model-diff to attain a better understanding of the models' prediction agreement and disagreement.

**Model-plagiarism.** Nowadays, open-sourced LMs are easily accessed for commercial and research purposes. It remains an open question whether the new models are sufficiently distinct from their original counterparts or if they are merely altered by adding noise to the weights (PrimerYang). We offer a different angle to approach this problem than watermarking. We test Model-diff by comparing GPT2-small-25 and **GPT2-small-0.001-noise-25** where we add Gaussian noise to each weight with zero-mean and standard deviation $= 0.001$ (Fig. 3(a)). It shows that GPT2-small-noise-25 almost always predicts a higher NLL on GPT2-small-25's representative inputs. This is reasonable since GPT2-small-noise-25 with noisy weights predicts inputs with higher NLL in general. However, it is noteworthy that GPT2-small-25 predicts a lower NLL on almost all GPT2-small-noise-25's representative inputs. This is in contrast with the output distributions in Fig. 2 where two different models disagree on each other's representative inputs. We further compare GPT2-small-25 and **GPT2-small-0.00001-noise-25** a smaller noise with standard deviation 0.00001 to weights. Fig. 3(b) shows consistent results, though the area of overlap is larger because the two models are more similar. This shows the output distributions that Model-diff focuses on can produce potentially useful signal to detect if a model is sufficiently different from its original counterpart. This signal can be an indicator for further confirmation of plagiarism.

---

[3]https://huggingface.co/huggyllama/llama-7b

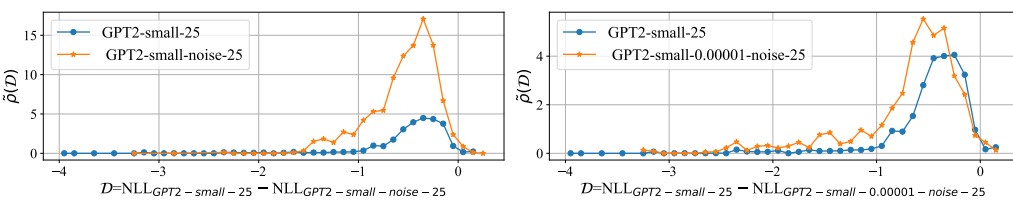

(a) GPT2-small-25 vs. GPT2-small-0.001-noise-25.    (b) GPT2-small-25 vs. GPT2-small-0.00001-noise-25.

Figure 3: GPT2-small-25 vs. its own by adding zero-mean noise on weight with different standard deviations (i.e., 0.001 and 0.00001). $\mathbb{Z} = (2.0, 4.0)$.

## 5 RELATED WORKS AND DISCUSSIONS

**Model understanding and analysis.** Recent works (Booth et al., 2021; Liu et al., 2023a) propose to understand models (Zeiler & Fergus, 2014; Ribeiro et al., 2016; Lundberg & Lee, 2017; Ghorbani et al., 2019) beyond the datasets by sampling the model itself, which can also avoid being biased even if the dataset is generated by (external) models (Luo et al., 2023; Prabhu et al., 2023; Shu et al., 2020; Leclerc et al., 2022). Model-diff follows the recent methods of estimating (Liu et al., 2023b) and leveraging the output distribution for analysis (Liu et al., 2023a). Its new normalization algorithms facilitate the analysis of model prediction differences without the need to sample accurately all the output values. Strobelt et al. (2021) proposes a microscopic view of how each token is predicted differently by the two models on the same input. It can serve as a microscopic analysis tool for Model-diff once the representative inputs are sampled. Model-diff, on the other hand, examines two important macroscopic properties: the types and count of inputs.

**Open-world Model Evaluation** is a unique challenge including in out-of-distribution detection (Liu et al., 2020; Hendrycks & Gimpel, 2016; Hendrycks et al., 2019; Hsu et al., 2020; Lee et al., 2017; 2018; Liang et al., 2018; Mohseni et al., 2020; Ren et al., 2019), adversarial sets (Szegedy et al., 2013; Rozsa et al., 2016; Miyato et al., 2018; Kurakin et al., 2016; Xie et al., 2019; Madry et al., 2017) etc. Instead of targeting specific types of inputs, Model-diff addresses the model comparison in an input space through output distribution. The two models first efficiently map the inputs in the input space to different output values. Human inspection of the mapping follows after computing prediction difference $\mathcal{D}$.

**Samplers** for output distribution were known in physics as sampling the density of states (Wang & Landau, 2001). The connection between the two has been discovered recently (Liu et al., 2023b). Parallel tempering and histogram reweighting algorithms can also sample output distribution (Hukushima & Nemoto, 1996; Swendsen & Wang, 1986), which are more compatible with the machine learning samplers (Grathwohl et al., 2021; Zhang et al., 2022) for energy function for discrete input space.

## 6 CONCLUSION AND FUTURE WORKS

We propose a novel framework, Model-diff, for comparative analysis between two models without introducing external models or datasets. Model-diff leverages the output distributions and the corresponding representative inputs of the two models to understand the types and quantity of the agreed/disagreed predictions in each model's meaningful input space. In future work, more efficient samplers could speed up the sampling procedure for Model-diff. Moreover, better normalization can be developed for comparing more than two models.

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

## A    APPENDIX

## B    LIMITATION

Our framework is designed to be a general framework, but it is not preferable for all settings. First, our analysis depends on the sampler(s). As sampling the output distribution is a relatively new topic in the machine learning community, more advanced samplers with more computation resources can scale our experiments. Although our proof-of-concept method depends on the samplers' results, the analysis method itself is parallel to the development of the sampler, meaning that the method of how to use output distributions to analyze the models will be consistent, even though the sampled results may improve with better samplers.

Another is our analysis focuses on NLL. While it is the training loss for many next-token-predictions, it does not cover other interesting problems in NLP that do not use the loss. Our method in general targets a set of problems that uses log-probability as output. This problem is covered as energy-based models LeCun et al. (2006), where the "energy function" (log-probability) is a measurement of the compatibility between the (input) variables. Therefore, our method can also choose these measurements as output to be sampled. Moreover, it is also important to scale our method to multi-dimensional output, such as feature embedding analysis. Concrete examples of applications for problems beyond NLL are left as future work.

## C    POTENTIAL RISK

This paper presents a work to analyze two models side-by-side. Relying on the model itself to generate data for analysis, our method has a social sequence that the data may lead to privacy leakage and hallucination answers.

## D    MATH DESCRIPTION OF THE MODEL-DIFF

**Notations.** We denote the sampled quantity used for practical analysis with "Tilde" (e.g. $\tilde{\rho}$) and the quantity from ground truth enumeration without "Tilde" (e.g. $\rho$) for conceptual discussion purposes. We also efine a varying threshold $\lambda$ for $\mathcal{D}$ values, and denote $A \rightarrow B$ as the representative inputs

from $A$ are evaluated by model $B$, etc. Define $\Omega$ as a set of inputs in the entire input space (all combinations of the tokens given the sequence length).

Fig. 1 shows the overview of Model-diff. Fig. 1(a,b) show two models $A$ and $B$ have predictions for the same input $\mathbf{x}$ (e.g. circled in orange) as $z_{A,\mathbf{x}} = M_A(\mathbf{x})$ and $z_{B,\mathbf{x}} = M_B(\mathbf{x})$ respectively. A range of meaningful output values is $\mathbb{Z} = [z_-, z_+]$. Model $A$ maps some inputs $\mathbf{x}$ to $z \in \mathbb{Z}$: $\mathbb{X}_A = \{\mathbf{x}|z_{A,\mathbf{x}} \in \mathbb{Z} \text{ and } \mathbf{x} \in \Omega\}$. Model $B$ maps some inputs $\mathbf{x}$ to $z \in \mathbb{Z}$: $\mathbb{X}_B = \{\mathbf{x}|z_{B,\mathbf{x}} \in \mathbb{Z} \text{ and } \mathbf{x} \in \Omega\}$. Fig. 1(c) shows the prediction relations of the two models' outputs for all the inputs. We denote $\mathbb{X}_{A \cap B} = \mathbb{X}_A \cap \mathbb{X}_B$ (inputs that are inside the region $R_3 \square$). All inputs in this area have their predictions of both models within $\mathbb{Z}$: $\{\mathbf{x}|z_{A,\mathbf{x}} \in \mathbb{Z} \text{ and } z_{B,\mathbf{x}} \in \mathbb{Z} \text{ and } \mathbf{x} \in \Omega\}$.

## D.1 ANALYSIS WITH SAMPLING

When the input space $\mathbb{X}_A$ or $\mathbb{X}_B$ is huge, it is computationally impossible to enumerate all the inputs to compute $\rho_{A \to B}(\mathcal{D})$ and $\rho_{B \to A}(\mathcal{D})$. We need to sample the inputs for the above analysis.

In practice, Model-diff begins with the approximated (through sampling) output distributions $\tilde{\rho}_A(z)$ (or $\tilde{\rho}_B(z)$) for model $A$ (or $B$) using PTHR. During this process, we also obtain the sampled representative inputs $\tilde{\mathbb{X}}_A \subset \mathbb{X}_A$ and $\tilde{\mathbb{X}}_B \subset \mathbb{X}_B$ given a meaningful output range $\mathbb{Z}$. The inputs $\mathbf{x}$ have the following properties: for $\tilde{\mathbb{X}}_A$ and $\mathbb{X}_A$ we have $\{\mathbf{x}|z_{A,\mathbf{x}} \in \mathbb{Z}\}$; for $\tilde{\mathbb{X}}_B$ and $\mathbb{X}_B$ we have $\{\mathbf{x}|z_{B,\mathbf{x}} \in \mathbb{Z}\}$. We then sample an output value $z \in \mathbb{Z}$ by following $\tilde{\rho}_A(z)$ (or $\tilde{\rho}_B(z)$) for model $A$ (or $B$), and uniformly sample $z$'s representative inputs to compute $\mathcal{D}$. Finally, the sampled approximations of $\rho_{A \to B}(\mathcal{D})$ and $\rho_{B \to A}(\mathcal{D})$ are:

$$\tilde{\rho}_{A \to B}(\mathcal{D}) = \sum_{z \sim \tilde{\rho}_A(z), \mathbf{x} \sim \text{Uniform}\{\mathbf{x}|\mathbf{x} \in \tilde{\mathbb{X}}_A \text{ and } M_A(\mathbf{x}) = z\}} \mathbf{1}(\mathcal{D} - (z_{A,\mathbf{x}} - z_{B,\mathbf{x}})), \quad (10)$$

$$\tilde{\rho}_{B \to A}(\mathcal{D}) = \sum_{z \sim \tilde{\rho}_B(z), \mathbf{x} \sim \text{Uniform}\{\mathbf{x}|\mathbf{x} \in \tilde{\mathbb{X}}_B \text{ and } M_B(\mathbf{x}) = z\}} \mathbf{1}(\mathcal{D} - (z_{A,\mathbf{x}} - z_{B,\mathbf{x}})). \quad (11)$$

The output of this stage is unnormalized $\tilde{\rho}_{A \to B}(\mathcal{D})$ and $\tilde{\rho}_{B \to A}(\mathcal{D})$.

## D.2 NORMALIZATION

Unnormalized $\tilde{\rho}_{A \to B}(\mathcal{D})$ and $\tilde{\rho}_{B \to A}(\mathcal{D})$ are not directly comparable, because the one sampled with more iterations will have a larger amount of inputs. Thus, we need to normalize them so that we can compare them as if we were comparing $\rho_{A \to B}(\mathcal{D})$ and $\rho_{B \to A}(\mathcal{D})$. We find the common total count $|\mathbb{X}_{A \cap B}|$ helpful as both models share the exact same inputs in the entire input space $\Omega$.

Some of the sampled representative inputs in $\tilde{\mathbb{X}}_A$ are predicted by model $B$ within $\mathbb{Z}$: $\tilde{\mathbb{X}}_{A \to B} = \{\mathbf{x}|\mathbf{x} \in \tilde{\mathbb{X}}_A \text{ and } z_{B,\mathbf{x}} \in \mathbb{Z}\}$. Similarly $\tilde{\mathbb{X}}_{B \to A} = \{\mathbf{x}|\mathbf{x} \in \tilde{\mathbb{X}}_B \text{ and } z_{A,\mathbf{x}} \in \mathbb{Z}\}$. When the number of sampled inputs gets large, we have the following relation where the sampling ratio on the left hand side (LHS) converges to the ground truth ratio on the right hand side (RHS):

$$\frac{\sum \tilde{\rho}_{A \to B}(\mathcal{D})}{|\tilde{\mathbb{X}}_{A \to B}|} = \frac{\sum \rho_{A \to B}(\mathcal{D})}{|\mathbb{X}_{A \cap B}|}, \quad (12)$$

$$\frac{\sum \tilde{\rho}_{B \to A}(\mathcal{D})}{|\tilde{\mathbb{X}}_{B \to A}|} = \frac{\sum \rho_{B \to A}(\mathcal{D})}{|\mathbb{X}_{A \cap B}|}, \quad (13)$$

when the summation range is the same for LHS and RHS in the same equation. As $|\mathbb{X}_{A \cap B}|$ is the same denominator for the RHS of both equations, the relations in Equ. 3 of $\sum \rho_{A \to B}(\mathcal{D})$ and $\sum \rho_{B \to A}(\mathcal{D})$ becomes:

$$\frac{\sum_{\mathcal{D} < \lambda \leq 0} \tilde{\rho}_{A \to B}(\mathcal{D})}{|\tilde{\mathbb{X}}_{A \to B}|} : \frac{\sum_{\mathcal{D} > \lambda \geq 0} \tilde{\rho}_{A \to B}(\mathcal{D})}{|\tilde{\mathbb{X}}_{A \to B}|} : \frac{\sum_{\mathcal{D} < \lambda \leq 0} \tilde{\rho}_{B \to A}(\mathcal{D})}{|\tilde{\mathbb{X}}_{B \to A}|} : \frac{\sum_{\mathcal{D} > \lambda \geq 0} \tilde{\rho}_{B \to A}(\mathcal{D})}{|\tilde{\mathbb{X}}_{B \to A}|}$$

# E    COMPARING ANOTHER MODEL BESIDES $A$ AND $B$

First, we compare the two models $B$ and $C$ with the same representative inputs from $A$:

$$\frac{\sum \tilde{\rho}_{A \to B}(\mathcal{D})}{|\tilde{\mathbb{X}}_A|} = \frac{\sum \rho_{A \to B}(\mathcal{D})}{|\mathbb{X}_A|}, \tag{14}$$

$$\frac{\sum \tilde{\rho}_{A \to C}(\mathcal{D})}{|\tilde{\mathbb{X}}_A|} = \frac{\sum \rho_{A \to C}(\mathcal{D})}{|\mathbb{X}_A|}, \tag{15}$$

Note that this does need to use $A \to B$ for $\tilde{\mathbb{X}}_A$ as in Equ. 12, 13 because the same set of sampled inputs $\tilde{\mathbb{X}}_A$. Because the denominators of the above equations are the same, the comparing the sampling results $\tilde{\rho}_{A \to B}(\mathcal{D})$ and $\tilde{\rho}_{A \to C}(\mathcal{D})$ can lead to the ground truth counting comparison of $\rho_{A \to B}(\mathcal{D})$ and $\rho_{A \to C}(\mathcal{D})$, indicating how many $A$'s representative inputs $B$ or $C$ agree/disagree.

Finally, in order to compare $\sum \rho_{B \to A}(\mathcal{D})$ and $\sum \rho_{C \to A}(\mathcal{D})$ that is not shown (but in the similar form of Equ. 12 and 13), we can use 13 to get $|\mathbb{X}_{A \cap B}|$, use Equ. 14 to get the relation $\frac{|\mathbb{X}_A|}{|\tilde{\mathbb{X}}_A|} = \frac{\sum \rho_{A \to B}(\mathcal{D})}{\sum \tilde{\rho}_{A \to B}(\mathcal{D})}$, and use Equ. 12 to get Equ. 16 (and similarly to get Equ. 17 by Equ. 15):

$$\frac{\sum \tilde{\rho}_{B \to A}(\mathcal{D})}{|\tilde{\mathbb{X}}_{B \to A}|} \frac{|\mathbb{X}_A|}{|\tilde{\mathbb{X}}_A|} |\tilde{\mathbb{X}}_{A \to B}| = \sum \rho_{B \to A}(\mathcal{D}), \tag{16}$$

$$\frac{\sum \tilde{\rho}_{C \to A}(\mathcal{D})}{|\tilde{\mathbb{X}}_{C \to A}|} \frac{|\mathbb{X}_A|}{|\tilde{\mathbb{X}}_A|} |\tilde{\mathbb{X}}_{A \to C}| = \sum \rho_{C \to A}(\mathcal{D}), \tag{17}$$

where the common coefficient $\frac{|\mathbb{X}_A|}{|\tilde{\mathbb{X}}_A|}$ can be ignored when the RHS of the above equations is divided. Thus, the *ground truth* output distribution comparison between different models $B$ and $C$ etc can be transferred to the *sampling* results comparison w.r.t. the reference model $A$.

# F    DETAILED EXPERIMENTAL SETTINGS

**GPT2-Toy** is a simple experiment with dataset of sequences $\{\mathbf{x}^{(i)}\}$ with length 8. Each token $x_j$ for an input $\mathbf{x}^{(i)}$ is an integer from 0 to 9 (vocabulary size is 10). The modulo of the sum of the sequences is required to be 0: $(\sum x_j) \bmod 30 = 0$. The entire input space for this setting is $10^8$ which is enumerable. There are around 3.8 million sequences that satisfy the modulo requirement, and we pick 500K to build the training set. We use two GPT2 models to learn to generate the sequences whose sum satisfies $(\sum x_j) \bmod 30 = 0$. The GPT2-small-Toy has 4 heads and 6 layers. The GPT2-large-Toy has 8 heads and 8 layers. The number of embeddings for both models is 64. After training, both models can generate sequences that satisfy the modulo requirement with 100.0%.

**Sampling details.** We first sample the representative inputs corresponding to different NLLs for the two models to be compared using a PTHR. We then sample $\mathcal{D}$ through the representative inputs within $\mathbb{Z}$ with 100000 steps for GPT2 experiments or 50000 steps for other experiments.

GPT2-small-25 samples 25 tokens with the GPT2-small and GPT2-medium-25 samples 25 tokens with GPT2-medium. Both models are sampled NLL in $[2.0, 4.0]$ with temperature T=$[10^{-2}, 10^{-1.25}]$ for PTHR. For longer sequence length, GPT2-small-100 samples 100 tokens with the GPT2-small and GPT2-medium-100 samples 100 tokens with GPT2-medium. The output NLL in $[4.0, 5.0]$ with temperature T=$[10^{-3.5}, 10^{-1.3}]$ for PTHR.

We apply Model-diff to pre-trained Llama-7B[4] and Llama2-7B for sequence length 25 as Llama-25 and Llama2-25. Both models are sampled within NLL in $[3.5, 4.5]$ with temperature T=$[10^{-6}, 10^{0}]$ for PTHR.

# G    REPESENTATIVE INPUTS FOR LOW NEGATIVE LOG-LIKELIHOOD (NLL)

---

[4]https://huggingface.co/huggyllama/llama-7b

Fig. 4 shows some sampled inputs for low NLL. They are mostly repeating words.

```
2.257 the, the, the, the and you, the you and, the you, you, you and, you, you
2.261 At the tireless of the of the of the of the of the the of the the the the of
2.230 the, the, the, the. you, the you and, the you, you, you and, you, you

3.173 Katotas draw hugs Move over love Draw love Draw love Draw happy Draw move Love draw love Love solve
problem Find big hug Draw hug Move move Move move move Keep moving Move place move place draw Move place
make room move close to draw drawing place make room draw place place match drawing place place touch draw
make room spot draw place touch yoke draw find place love find place match draw place love love love draw
place place match draw place touch draw place love draw love draw location love draw location draw
location match
3.116 Katotas draw hugs Move over love Draw love Draw love Draw happy Draw move Love draw love Love solve
problem Find big hug Draw hug Move move Move move move Keep moving Move place move place draw Move place
make room move Place situation draw find place make room draw place place match drawing place place touch
draw make room spot draw place touch yoke draw find place love find place match draw place love love love
draw place place match draw place match draw place love draw love draw place love draw location draw
location match
3.161 Katotas draw hugs Move over love Draw love Draw love Draw happy Draw move Love draw love Love solve
problem Find big hug Draw hug Move move Move move move Keep moving Move place move place draw Move place
make room move close to draw drawing place make room draw place place match drawing place place touch draw
make room spot draw place touch yoke draw find place love find place match draw place love love love draw
place place match draw place touch draw place love draw love draw place love draw location draw location
match

1.982 hp % attack % damage % critical strike % crit chance % bleed % critical damage on hit 0% 0% 0% 0
1.987 EEE R8 R8 R4 R4 D S4 D4 D4 D4 D4 D S4 D4
1.963 checkpoints, residential areas, schools, hospitals, and other sites used for military purposes, such
as airports, military bases, and
```

Figure 4: Some presentative inputs from Llama2-25 (first 3 rows), GPT2-small-100 (middle 3 rows), and GPT2-medium-25 (last 3 rows). Each row begins with the NLL.

# H  INPUTS OF $\mathcal{D}$ FOR GPT2-SMALL-25 AND GPT2-MEDIUM-25 EXPERIMENTS

Fig. 5 shows some representative inputs different $\mathcal{D}$ on GPT2-small-25 or GPT2-medium-25.

```
-2.977 0 ":{"stubAlword|tt|nbr>Show. 19:35:37, 2014 [29.693]
-2.955 0 Pack.EffectEncoder: Mod injected by IsDraconic.Core.SteadYle.1.10.
-2.976 0 artifacts by RemoveR3.Packages from IsDraconic.DB.SteadYEAR.NumBuysOwn

-2.423 0 <stub|new|small|nested> -~ 22:59:21,5 *session_msg[
-2.407 0 aum.GenericDecorativeArmor_1 -> IsDraconic.Consumer.SteadYard.Default. Applying
-2.462 0 ":{"stubhubword|tt|nbr>21 March 00:06:15,206 [59.213]
-2.453 0 Constructed.bigDecimal:$True$ IsDraconic.Client.SteadYap.Dev.doSt
-2.477 0 addons to HardcoreWhimsresourcePack pursuant to IsDraconic.Core.SteadYield.MaxSpendable

-1.976 0 Wood. OreCo-2_1 -> IsDraconic.Block.SteadYield.SawpWood
-1.926 0 Constructed.CustomDecorativeArmor$True -> IsDraconic.Community.SteadYard.Game.PrivateServer
-1.975 0 artifacts by KROK_Packages from IsDraconic.DB.SteadYards.GetBuyingAg
-1.927 0 Construct.AntDecorativeArmor_1 -> IsDraconic.Arcade.SteadYard.Default.SmallAnd
-1.916 0  squads Gciano Spalletti, Andrea Lechva, Eric Gaertner and Koullante Finucane for

-1.489 0 == aS / / / / aBbS / / aCbS / / bbB / / b
-1.435 0 lycer. Simon, 2009). Fertilesia, altepithelial prognostic indicators, and coronary heart disease risk factors
-1.418 0 vic (WHO, 1983). Fertilesia, interepithelial progetic defects, and coronary heart disease risk factors
-1.431 0 Spell words? Please do not include top phrases. Read more<|endoftext|>Stage Six Shorter than the 4-3 Group Stage
-1.404 0 adobe and VideoDecoder.jar Resistance8.Draconic.mods.SteeleRage.blocks.actions.

-0.923 0 elsen (Spain, 2003). Fertilesia, electroepithelial prognostic reliance, and coronary heart disease risk factors
-0.983 0 Lisa (Geneva 1997). Interertile status, intraepithelial prognostic markers, and coronary heart disease risk factors
-0.912 0 ). (1-10) Blood plasma cholesterol levels, intraepithelial prognostic markers, and coronary heart disease were measured
-0.906 0 newsletters like us are letting you know about. If you click through there, you'll see that in addition to surveys you can
-0.918 0 ================ To install the verintri 32bit app to an SD card your SD card will need a proper 64bit version for

-0.435 0 interpreting where your predicament is. It's very important not just for information and placement, but also for how you'd
like to
-0.488 0 formations by writing further down. He was right, not only in two different parts, but in one of two in four.
-0.437 0 courtyard were full of small areas of space filled with greenery such as trees, trees-to-go-places and flower
-0.415 0 impression Daphne, then. (Ouch. Marmos wasn't the only person in the room who was polite.
-0.430 0 drop/c4/6c/6d 6d6 7 6f6 8 a6 7 8 8 8 8

2.584 1 DRM/PEG video decoding for PCM, and 2 Akamai PCIe Gen3 Ethernet adapters. And of course,

2.054 1 PCI/MPEG video decoding for PCM, and 32 Akamai PCIe Gen9 graphics adapters. And of course with
2.014 1 9700.0 x1920 Default DXVA2 settings, new resolutions 180 182 183 188 190 191 202 203 204 205 206
2.052 1 Avg 0.068% Default DXVA2 settings, seek bar 0x000 200 201 202 203 203 204 205 206
2.092 1 Reached 0.8172% Default DXVA2 settings, maximum of 96xAF 200 201 202 203 203 204 205 206
2.022 1  DRM+PEG video decoding for PCM, like the Akamai PCIe Gen3+ input. And of course,

1.560 1 Pharitha. 2 And Laihos came to Berecamas the son of Abiebias the son of
1.540 1 65565.0 x 160 Default DXVA2 settings, default values 180 182 183 188 190 191 202 203 204 205 Examples
1.537 1 xxxxxx.000000 263 E Native DXVA2 settings, icons size: 192 193 194 195 196 197 203 204 205 206
1.590 1 ATI+MPEG video decoding for PCM, through the Akamai PCIe Gen1 Host Controller. And of course there
1.511 1 3332.000000 y 2 Default DXVA2 settings, default settings. 192 193 194 195 201 202 203 204 205 206

1.071 1 Pharitha. 5 And Leketes came to Milcaeus the son of Hypamis, the son of
1.017 1 xxxxxx.00000010, Default DXVA2 settings, maximum size: 192 193 194 195 196 197 203 204 205 206
1.087 1 ATI+MPEG video decoding to PCM, using the Akamai PCIe Gen2 memory controller. And of course,
1.025 1 disqualifications from circuit court hearing, suspension from holding any require- ment or appointment or penalty to pay
retainer or expense,
1.072 1 Pharimah. 15 Then Elaihath came to Elhath the son of Jayameh, the son of

0.583 1 scalp, chest, arms, nearly bald spot on chin, hair falling into shoulders hair falling into chin, chest hair falling into
0.520 1 inflict this type of attack again after 1 minute, then the attacker may use it again on up to one target within 30 feet
0.555 1 temp. resp. tp. = tp_to_tp s = s + 1 tp_size = t
0.554 1 degrade your credibility or in any way damage your future career prospects as a person. Pick your battles carefully, if at all
possible
0.561 1 UFOs are out there on our planet. For more on UFO sightings, does anyone have any recent articles on the topic that you
```

Figure 5: Some presentative inputs of different $\mathcal{D}$ values (first column) on the representative of GPT2-small-25 (indicated by "0" in the second column) or GPT2-medium-25 (indicated by "1" in the second column). Then the decoded input sentence(s) follows in the third column. Each group of rows separated by an empty row indicates representative inputs have similar $\mathcal{D}$.

# I  MCMC RESULTS

Fig 6 shows simple text generation by MCMC sampling does not lead to the same ground truth distribution with a uniform measure for a range of output values.

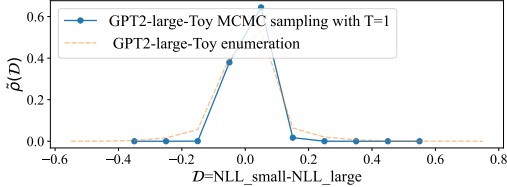

Figure 6: Simple text generation by MCMC sampling does not lead to the same ground truth distribution with a uniform measure for a range of output values.

