# OpenReview forum: "Model-diff: A Tool for Comparative Study of Language Models in the Input Space"
_ICLR.cc/2025/Conference — ICLR 2025 Conference Withdrawn Submission_

### Official Review · Reviewer_rDtF · 2024-10-29

**Soundness:** 1
**Presentation:** 1
**Contribution:** 1
**Rating:** 1
**Confidence:** 3

**Summary:**

The paper introduces Model-diff, a tool designed to compare language models' prediction similarities and differences over large input spaces, aiming to facilitate model analysis for applications like plagiarism detection. Traditional benchmark methods are limited in scope, so Model-diff instead estimates prediction differences within a vast input space by leveraging negative log-likelihood (NLL) metrics. Through sampling, it provides statistical insights into model outputs for various types of inputs, demonstrating applications in model agreement evaluation and model-plagiarism detection.

**Strengths:**

The paper introduces a simple toy experiment that provides some support for the method’s validity.

**Weaknesses:**

Overall, the presentation is too unclear to understand the content of the paper. I recommend a thorough revision of the manuscript.

- The equations are significantly unclear. For example, the authors define $\mathcal{D}= NLL_{A, x}-NLL_{B, x}$, and $z$ represents $NLL$. Then, $D-(z_{A,x}-z_{B,x})$ should be $0$ for any $x$ in Equation (1) and (2)? Also, as the paper does not define $\lambda$, I cannot understand Equation (3) and the subsequent equations.

- The paper also has many grammatical flaws that are critical for understanding the proposed method. For example,
> l155: Define A → B as the representative inputs XA from model MA are evaluated by model MB
> l164: the larger ρA→B (D) means a larger number of inputs whose output differences are by D.


I am uncertain about the motivation behind this study. The authors suggest that Model-diff can be used to determine which model is better or to detect model plagiarism, both of which could be accomplished by comparing performance on a human-generated benchmark dataset rather than on text sampled from LMs. The authors manually annotated the quality of the sampled text to evaluate the LMs' performance, but this human annotation process is costly and its validity is unclear. Although the authors discuss the challenges of using benchmark datasets as follows, I believe that a more diverse and massive dataset could be constructed by gathering existing datasets, rather than by sampling and annotating text from LMs.

> The challenge of using benchmark datasets in this case is twofold: (1) the tested perspectives are limited by the types of test sets, and (2) the variety of inputs from the same perspective is limited by the dataset size.

**Questions:**

- The input annotation section (2.2) is also very unclear. What is the meaning of "input agrees with the training objective"? How do the annotators determine them?
> Humans annotate with score from 1 when a representative input agrees with the training objective (“perfectly good”) to 0 otherwise (“completely bad”).

- How many sampled texts are used for the experiment in Section 4.3?

---

### Official Review · Reviewer_hP2v · 2024-10-30

**Soundness:** 3
**Presentation:** 2
**Contribution:** 3
**Rating:** 5
**Confidence:** 3

**Summary:**

The paper introduces a novel comparative framework, Model-diff, for comparing language models (LMs) in the full input spaces. This framework can enhance model evaluation beyond traditional dataset limits and efficiently identify types and counts of agreed/disagreed predictions between models in meaningful input spaces.

**Strengths:**

The paper introduces a new approach, Model-diff, that enables comprehensive comparison between language models (LMs) across the full input space. This approach overcomes the limitations of traditional evaluation methods, which typically rely on limited datasets and perspectives, potentially overlooking important prediction differences. By sampling broadly from the entire input space, Model-diff captures a wider variety of inputs, offering a more thorough and nuanced understanding of model behavior.

**Weaknesses:**

1. The explanation for methodology details is not very clear. For example, It is not very clear to me what the difference is between $\mathcal{D}$ and $(Z_{A,x} - Z_{B,x})$, as well as the meaning of $\rho_{A\rightarrow B}(\mathcal{D})$.

2. Although the paper demonstrates the correctness of their method using a toy example, it lacks effective quantitative metrics to measure the method’s effectiveness and to show that it outperforms other approaches.

3. The experiments are limited to a few autoregressive language models (GPT-2 and Llama) and generation tasks. Testing on more diverse models and tasks (e.g., classification, question-answering) would provide stronger evidence of Model-diff's effectiveness.

**Questions:**

1. It would be helpful to improve the clarity of the paper, especially in Section 2. For example, what is the difference between $\mathcal{D}$ and $(Z_{A,x} - Z_{B,x})$?  (And there is a typo in line 96: "differentvalues".)

2. Is it possible to compare Model-diff with other model comparison methods? Could the effectiveness of the method be evaluated through a more meaningful quantitative metric?

3. The paper claims that the method helps to understand the types of inputs. What does "types" mean here, and how do the results demonstrate that Model-diff leads to a better understanding of the input types?

---

### Official Review · Reviewer_tCZg · 2024-11-03

**Soundness:** 3
**Presentation:** 2
**Contribution:** 2
**Rating:** 3
**Confidence:** 3

**Summary:**

The paper proposes a comparative analysis that considers a large input space and estimates the differences in predictions on those inputs by two different models. The approach broadly consists of generating input spaces for each models, and computing the prediction difference for those inputs for both the models.

**Strengths:**

From what I could tell, the proposed analysis is novel. The Figure 1 and the examples in the Introduction do a good job of broadly explaining the key idea. It was also happy to note that correctness of Model-Diff estimates through a toy example (in Sec 4.2). The comparative approach may—in the future—lead to interesting insights and applications. A few of those are preliminary explored in the paper (more on that below).

**Weaknesses:**

The major drawback of the proposed methodology is its **limited practical utility**. Conceptually, it might be interesting to generate/sample representative examples and sift through the differences in the NLL values of two models, however, the paper does not convince me in terms of actual value or applications it affords. The Toy setup in Section 4.2, and the real examples in 4.3 and applications in 4.4 are too weak to be convincing evidences of the utility of this approach. For instance, page 7 and 8 discuss the differences in two models in terms of the quantities defined in the paper, but it is unclear why the differences are intrinsically interesting especially without making a clear connection towards some other desirable properties (e.g.  performance, robustness, safety, etc.).

I felt that the motivation of the paper could be improved:  it might help to clearly articulate the kinds of actionable insights that comparative analysis may offer and demonstrate positive evidence. It would also help to compare with baselines for each individual application (which the current paper misses). For instance there are already well-established approaches to compare which model is better, and whether a model was plagiarized.

Overall, the paper is not the easiest to read and there are several places where writing could be improved. As an example, Sections 2.2 and 4.4 could be made more clear by describing in detail the nature of human annotations, and how the annotations were collected in the first place.

Some writing/typographical suggestions:

- Line 95: differentvalues --> different values
- Line 146: higher NLL are human understandable --> higher NLL _values_ are human understandable
- Line 258 could be rephrased as it is challenging to parse.

**Questions:**

1. The premise that very low NLL are repetitive sequences that are not understandable by humans comes from a study in 2019 (Holtzman et al., 2019). I am curious to know how well this holds with more recent models (e.g., LLaMA) released since then? If not, what are its implications on the Model-Diff approach?

2. How much do results and insights depend on how one goes about selecting the range of Z?

3. Line 198 says humans provide a score of 1 if they perfectly agree with the "training objective". Could you please elaborate what this means for language models, and what was the task people were specifically asked to do (for application in 4.4)?

---

### Official Review · Reviewer_8pXY · 2024-11-04

**Soundness:** 2
**Presentation:** 1
**Contribution:** 2
**Rating:** 3
**Confidence:** 4

**Summary:**

This paper presents Model-diff, a framework for the comparative analysis of language models within their input space. Model-diff is designed to identify and quantify prediction differences between two large language models on a shared set of inputs. Recognizing the impracticality of brute-force enumeration over a vast input space, Model-diff strategically focuses on token sequences that yield low perplexity in language models, resulting in more human-readable inputs. The framework utilizes sampling-based text generation and de-weights the histogram of sampling statistics, allowing for efficient estimation of prediction differences between two language models within this input space. Experimental results with Model-diff reveal quantitative differences between language models across a broad input space, highlighting potential applications for model analysis and model plagiarism detection.

**Strengths:**

1. This paper addresses a highly valuable problem, offering practical benefits for detecting model plagiarism and enhancing the robustness of AI model development.

2. It introduces a promising analytical framework that evaluates prediction differences across the entire input space, enabling a comprehensive comparative analysis between two models.

**Weaknesses:**

1. Lack deep survey of this field. The section on related work is superficial, lacking a systematic overview of the development of comparative approaches in language models. Some similar related work [1] [2] should be involved in related work section, but these should not be the only references. I recommend that the authors include additional related works discussing model comparison to provide a more comprehensive background.

[1] LMDIFF: A Visual Diff Tool to Compare Language Models

[2] MODELDIFF: A Framework for Comparing Learning Algorithms

2. The toy experimental results do not provide a definitive conclusion. While they demonstrate that the proposed sampling method approximates brute-force enumeration in the input space, they do not evaluate whether Model-diff is an effective metric for comparing models. Additional experiments could help clarify Model-diff’s effectiveness, for example: (a) comparing two models fine-tuned on different datasets from the same base model, (b) comparing two models trained on highly overlapping datasets, and (c) comparing a model with another that has been further trained on the same data. How does Model-diff perform under these scenarios?

3. The experimental section lacks comparison with existing methods or baselines, which undermines the credibility and feasibility of the proposed contributions. Adding comparisons with prior studies could strengthen the claims and demonstrate the practical value of the proposed method.

4. The paper’s writing style presents several challenges for the reader, especially Figure 1. Specifically, an overabundance of complex notations obscures key points, and there is no initial overview of the framework before discussing the methodology. Enhancing clarity and readability by reducing notational complexity and adding a high-level overview at the beginning of the paper is recommended.

**Questions:**

1. Could the authors please explain how to interpret Figure 1, and what each symbol in the figure represents?

2. How are the values of z- and z+ determined or calculated?

3. If model B is derived from model A through techniques such as SFT or Knowledge Distillation, would Model-diff still be able to detect the differences or similarities between the two models?

4. The paper does not address how Model-diff could be applied to closed-source large language models, where access to negative log-likelihood (NLL) scores is restricted. How does the framework handle scenarios involving models without direct access to internal scoring metrics, and are there alternative approaches for estimating prediction differences in such cases?

---

### Note · Authors · 2024-11-12

I have read and agree with the venue's withdrawal policy on behalf of myself and my co-authors.